# Generative Modeling by Estimating Gradients of the Data Distribution

**Yang Song**
Stanford University
yangsong@cs.stanford.edu

**Stefano Ermon**
Stanford University
ermon@cs.stanford.edu

## Abstract

We introduce a new generative model where samples are produced via Langevin dynamics using gradients of the data distribution estimated with score matching. Because gradients can be ill-defined and hard to estimate when the data resides on low-dimensional manifolds, we perturb the data with different levels of Gaussian noise, and jointly estimate the corresponding scores, *i.e.*, the vector fields of gradients of the perturbed data distribution for all noise levels. For sampling, we propose an annealed Langevin dynamics where we use gradients corresponding to gradually decreasing noise levels as the sampling process gets closer to the data manifold. Our framework allows flexible model architectures, requires no sampling during training or the use of adversarial methods, and provides a learning objective that can be used for principled model comparisons. Our models produce samples comparable to GANs on MNIST, CelebA and CIFAR-10 datasets, achieving a new state-of-the-art inception score of 8.87 on CIFAR-10. Additionally, we demonstrate that our models learn effective representations via image inpainting experiments.

## 1 Introduction

Generative models have many applications in machine learning. To list a few, they have been used to generate high-fidelity images [26, 6], synthesize realistic speech and music fragments [58], improve the performance of semi-supervised learning [28, 10], detect adversarial examples and other anomalous data [54], imitation learning [22], and explore promising states in reinforcement learning [41]. Recent progress is mainly driven by two approaches: likelihood-based methods [17, 29, 11, 60] and generative adversarial networks (GAN [15]). The former uses log-likelihood (or a suitable surrogate) as the training objective, while the latter uses adversarial training to minimize $f$-divergences [40] or integral probability metrics [2, 55] between model and data distributions.

Although likelihood-based models and GANs have achieved great success, they have some intrinsic limitations. For example, likelihood-based models either have to use specialized architectures to build a normalized probability model (*e.g.*, autoregressive models, flow models), or use surrogate losses (*e.g.*, the evidence lower bound used in variational auto-encoders [29], contrastive divergence in energy-based models [21]) for training. GANs avoid some of the limitations of likelihood-based models, but their training can be unstable due to the adversarial training procedure. In addition, the GAN objective is not suitable for evaluating and comparing different GAN models. While other objectives exist for generative modeling, such as noise contrastive estimation [19] and minimum probability flow [50], these methods typically only work well for low-dimensional data.

In this paper, we explore a new principle for generative modeling based on estimating and sampling from the *(Stein) score* [33] of the logarithmic data density, which is the gradient of the log-density function at the input data point. This is a vector field pointing in the direction where the log data density grows the most. We use a neural network trained with score matching [24] to learn this vector field from data. We then produce samples using Langevin dynamics, which approximately

works by gradually moving a random initial sample to high density regions along the (estimated) vector field of scores. However, there are two main challenges with this approach. First, if the data distribution is supported on a low dimensional manifold—as it is often assumed for many real world datasets—the score will be undefined in the ambient space, and score matching will fail to provide a consistent score estimator. Second, the scarcity of training data in low data density regions, *e.g.*, far from the manifold, hinders the accuracy of score estimation and slows down the mixing of Langevin dynamics sampling. Since Langevin dynamics will often be initialized in low-density regions of the data distribution, inaccurate score estimation in these regions will negatively affect the sampling process. Moreover, mixing can be difficult because of the need of traversing low density regions to transition between modes of the distribution.

To tackle these two challenges, we propose to *perturb the data with random Gaussian noise of various magnitudes*. Adding random noise ensures the resulting distribution does not collapse to a low dimensional manifold. Large noise levels will produce samples in low density regions of the original (unperturbed) data distribution, thus improving score estimation. Crucially, we train a single score network conditioned on the noise level and estimate the scores at all noise magnitudes. We then propose *an annealed version of Langevin dynamics*, where we initially use scores corresponding to the highest noise level, and gradually anneal down the noise level until it is small enough to be indistinguishable from the original data distribution. Our sampling strategy is inspired by simulated annealing [30, 37] which heuristically improves optimization for multimodal landscapes.

Our approach has several desirable properties. First, our objective is tractable for almost all parameterizations of the score networks without the need of special constraints or architectures, and can be optimized without adversarial training, MCMC sampling, or other approximations during training. The objective can also be used to quantitatively compare different models on the same dataset. Experimentally, we demonstrate the efficacy of our approach on MNIST, CelebA [34], and CIFAR-10 [31]. We show that the samples look comparable to those generated from modern likelihood-based models and GANs. On CIFAR-10, our model sets the new state-of-the-art inception score of 8.87 for unconditional generative models, and achieves a competitive FID score of 25.32. We show that the model learns meaningful representations of the data by image inpainting experiments.

## 2 Score-based generative modeling

Suppose our dataset consists of i.i.d. samples $\{\mathbf{x}_i \in \mathbb{R}^D\}_{i=1}^N$ from an unknown data distribution $p_{\text{data}}(\mathbf{x})$. We define the *score* of a probability density $p(\mathbf{x})$ to be $\nabla_{\mathbf{x}} \log p(\mathbf{x})$. The *score network* $\mathbf{s}_{\boldsymbol{\theta}} : \mathbb{R}^D \to \mathbb{R}^D$ is a neural network parameterized by $\boldsymbol{\theta}$, which will be trained to approximate the score of $p_{\text{data}}(\mathbf{x})$. The goal of generative modeling is to use the dataset to learn a model for generating new samples from $p_{\text{data}}(\mathbf{x})$. The framework of score-based generative modeling has two ingredients: score matching and Langevin dynamics.

### 2.1 Score matching for score estimation

Score matching [24] is originally designed for learning non-normalized statistical models based on i.i.d. samples from an unknown data distribution. Following [53], we repurpose it for score estimation. Using score matching, we can directly train a score network $\mathbf{s}_{\boldsymbol{\theta}}(\mathbf{x})$ to estimate $\nabla_{\mathbf{x}} \log p_{\text{data}}(\mathbf{x})$ without training a model to estimate $p_{\text{data}}(\mathbf{x})$ first. Different from the typical usage of score matching, we opt not to use the gradient of an energy-based model as the score network to avoid extra computation due to higher-order gradients. The objective minimizes $\frac{1}{2} \mathbb{E}_{p_{\text{data}}}[\|\mathbf{s}_{\boldsymbol{\theta}}(\mathbf{x}) - \nabla_{\mathbf{x}} \log p_{\text{data}}(\mathbf{x})\|_2^2]$, which can be shown equivalent to the following up to a constant

$$\mathbb{E}_{p_{\text{data}}(\mathbf{x})}\left[\operatorname{tr}(\nabla_{\mathbf{x}}\mathbf{s}_{\boldsymbol{\theta}}(\mathbf{x})) + \frac{1}{2}\|\mathbf{s}_{\boldsymbol{\theta}}(\mathbf{x})\|_2^2\right], \tag{1}$$

where $\nabla_{\mathbf{x}}\mathbf{s}_{\boldsymbol{\theta}}(\mathbf{x})$ denotes the Jacobian of $\mathbf{s}_{\boldsymbol{\theta}}(\mathbf{x})$. As shown in [53], under some regularity conditions the minimizer of Eq. (3) (denoted as $\mathbf{s}_{\boldsymbol{\theta}^*}(\mathbf{x})$) satisfies $\mathbf{s}_{\boldsymbol{\theta}^*}(\mathbf{x}) = \nabla_{\mathbf{x}} \log p_{\text{data}}(\mathbf{x})$ almost surely. In practice, the expectation over $p_{\text{data}}(\mathbf{x})$ in Eq. (1) can be quickly estimated using data samples. However, score matching is not scalable to deep networks and high dimensional data [53] due to the computation of $\operatorname{tr}(\nabla_{\mathbf{x}}\mathbf{s}_{\boldsymbol{\theta}}(\mathbf{x}))$. Below we discuss two popular methods for large scale score matching.

**Denoising score matching** Denoising score matching [61] is a variant of score matching that completely circumvents $\operatorname{tr}(\nabla_{\mathbf{x}}\mathbf{s}_{\boldsymbol{\theta}}(\mathbf{x}))$. It first perturbs the data point $\mathbf{x}$ with a pre-specified noise

distribution $q_\sigma(\tilde{\mathbf{x}} \mid \mathbf{x})$ and then employs score matching to estimate the score of the perturbed data distribution $q_\sigma(\tilde{\mathbf{x}}) \triangleq \int q_\sigma(\tilde{\mathbf{x}} \mid \mathbf{x}) p_{\text{data}}(\mathbf{x}) \mathrm{d}\mathbf{x}$. The objective was proved equivalent to the following:

$$\frac{1}{2} \mathbb{E}_{q_\sigma(\tilde{\mathbf{x}}|\mathbf{x}) p_{\text{data}}(\mathbf{x})} [\|\mathbf{s}_\theta(\tilde{\mathbf{x}}) - \nabla_{\tilde{\mathbf{x}}} \log q_\sigma(\tilde{\mathbf{x}} \mid \mathbf{x})\|_2^2]. \tag{2}$$

As shown in [61], the optimal score network (denoted as $\mathbf{s}_{\theta^*}(\mathbf{x})$) that minimizes Eq. (2) satisfies $\mathbf{s}_{\theta^*}(\mathbf{x}) = \nabla_{\mathbf{x}} \log q_\sigma(\mathbf{x})$ almost surely. However, $\mathbf{s}_{\theta^*}(\mathbf{x}) = \nabla_{\mathbf{x}} \log q_\sigma(\mathbf{x}) \approx \nabla_{\mathbf{x}} \log p_{\text{data}}(\mathbf{x})$ is true only when the noise is small enough such that $q_\sigma(\mathbf{x}) \approx p_{\text{data}}(\mathbf{x})$.

**Sliced score matching** Sliced score matching [53] uses random projections to approximate $\text{tr}(\nabla_{\mathbf{x}} \mathbf{s}_\theta(\mathbf{x}))$ in score matching. The objective is

$$\mathbb{E}_{p_{\mathbf{v}}} \mathbb{E}_{p_{\text{data}}} \left[ \mathbf{v}^\mathsf{T} \nabla_{\mathbf{x}} \mathbf{s}_\theta(\mathbf{x}) \mathbf{v} + \frac{1}{2} \|\mathbf{s}_\theta(\mathbf{x})\|_2^2 \right], \tag{3}$$

where $p_{\mathbf{v}}$ is a simple distribution of random vectors, *e.g.*, the multivariate standard normal. As shown in [53], the term $\mathbf{v}^\mathsf{T} \nabla_{\mathbf{x}} \mathbf{s}_\theta(\mathbf{x}) \mathbf{v}$ can be efficiently computed by forward mode auto-differentiation. Unlike denoising score matching which estimates the scores of *perturbed* data, sliced score matching provides score estimation for the original *unperturbed* data distribution, but requires around four times more computations due to the forward mode auto-differentiation.

## 2.2 Sampling with Langevin dynamics

Langevin dynamics can produce samples from a probability density $p(\mathbf{x})$ using only the score function $\nabla_{\mathbf{x}} \log p(\mathbf{x})$. Given a fixed step size $\epsilon > 0$, and an initial value $\tilde{\mathbf{x}}_0 \sim \pi(\mathbf{x})$ with $\pi$ being a prior distribution, the Langevin method recursively computes the following

$$\tilde{\mathbf{x}}_t = \tilde{\mathbf{x}}_{t-1} + \frac{\epsilon}{2} \nabla_{\mathbf{x}} \log p(\tilde{\mathbf{x}}_{t-1}) + \sqrt{\epsilon}\, \mathbf{z}_t, \tag{4}$$

where $\mathbf{z}_t \sim \mathcal{N}(0, I)$. The distribution of $\tilde{\mathbf{x}}_T$ equals $p(\mathbf{x})$ when $\epsilon \to 0$ and $T \to \infty$, in which case $\tilde{\mathbf{x}}_T$ becomes an exact sample from $p(\mathbf{x})$ under some regularity conditions [62]. When $\epsilon > 0$ and $T < \infty$, a Metropolis-Hastings update is needed to correct the error of Eq. (4), but it can often be ignored in practice [9, 12, 39]. In this work, we assume this error is negligible when $\epsilon$ is small and $T$ is large.

Note that sampling from Eq. (4) only requires the score function $\nabla_{\mathbf{x}} \log p(\mathbf{x})$. Therefore, in order to obtain samples from $p_{\text{data}}(\mathbf{x})$, we can first train our score network such that $\mathbf{s}_\theta(\mathbf{x}) \approx \nabla_{\mathbf{x}} \log p_{\text{data}}(\mathbf{x})$ and then approximately obtain samples with Langevin dynamics using $\mathbf{s}_\theta(\mathbf{x})$. This is the key idea of our framework of *score-based generative modeling*.

# 3 Challenges of score-based generative modeling

In this section, we analyze more closely the idea of score-based generative modeling. We argue that there are two major obstacles that prevent a naïve application of this idea.

## 3.1 The manifold hypothesis

The manifold hypothesis states that data in the real world tend to concentrate on low dimensional manifolds embedded in a high dimensional space (a.k.a., the ambient space). This hypothesis empirically holds for many datasets, and has become the foundation of manifold learning [3, 47]. Under the manifold hypothesis, score-based generative models will face two key difficulties. First, since the score $\nabla_{\mathbf{x}} \log p_{\text{data}}(\mathbf{x})$ is a gradient taken in the *ambient space*, it is undefined when $\mathbf{x}$ is confined to a low dimensional

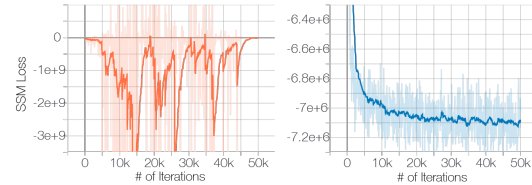

Figure 1: **Left**: Sliced score matching (SSM) loss w.r.t. iterations. No noise is added to data. **Right**: Same but data are perturbed with $\mathcal{N}(0, 0.0001)$.

manifold. Second, the score matching objective Eq. (1) provides a consistent score estimator only when the support of the data distribution is the whole space (*cf*., Theorem 2 in [24]), and will be inconsistent when the data reside on a low-dimensional manifold.

The negative effect of the manifold hypothesis on score estimation can be seen clearly from Fig. 1, where we train a ResNet (details in Appendix B.1) to estimate the data score on CIFAR-10. For fast training and faithful estimation of the data scores, we use the sliced score matching objective (Eq. (3)). As Fig. 1 (left) shows, when trained on the original CIFAR-10 images, the sliced score matching loss first decreases and then fluctuates irregularly. In contrast, if we perturb the data with a small Gaussian noise (such that the perturbed data distribution has full support over $\mathbb{R}^D$), the loss curve will converge (right panel). Note that the Gaussian noise $\mathcal{N}(0, 0.0001)$ we impose is very small for images with pixel values in the range $[0, 1]$, and is almost indistinguishable to human eyes.

## 3.2 Low data density regions

The scarcity of data in low density regions can cause difficulties for both score estimation with score matching and MCMC sampling with Langevin dynamics.

### 3.2.1 Inaccurate score estimation with score matching

In regions of low data density, score matching may not have enough evidence to estimate score functions accurately, due to the lack of data samples. To see this, recall from Section 2.1 that score matching minimizes the expected squared error of the score estimates, *i.e.*, $\frac{1}{2}\mathbb{E}_{p_{\text{data}}}[\|\mathbf{s}_{\boldsymbol{\theta}}(\mathbf{x}) - \nabla_{\mathbf{x}}\log p_{\text{data}}(\mathbf{x})\|_2^2]$. In practice, the expectation w.r.t. the data distribution is always estimated using i.i.d. samples $\{\mathbf{x}_i\}_{i=1}^N \overset{\text{i.i.d.}}{\sim} p_{\text{data}}(\mathbf{x})$. Consider any region $\mathcal{R} \subset \mathbb{R}^D$ such that $p_{\text{data}}(\mathcal{R}) \approx 0$. In most cases $\{\mathbf{x}_i\}_{i=1}^N \cap \mathcal{R} = \varnothing$, and score matching will not have sufficient data samples to estimate $\nabla_{\mathbf{x}}\log p_{\text{data}}(\mathbf{x})$ accurately for $\mathbf{x} \in \mathcal{R}$.

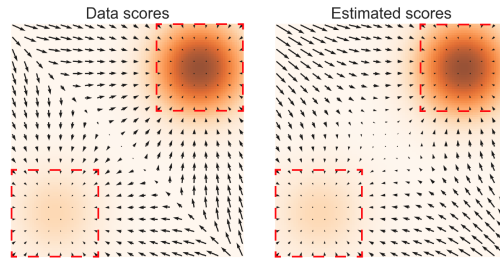

Figure 2: **Left**: $\nabla_{\mathbf{x}}\log p_{\text{data}}(\mathbf{x})$; **Right**: $\mathbf{s}_{\boldsymbol{\theta}}(\mathbf{x})$. The data density $p_{\text{data}}(\mathbf{x})$ is encoded using an orange colormap: darker color implies higher density. Red rectangles highlight regions where $\nabla_{\mathbf{x}}\log p_{\text{data}}(\mathbf{x}) \approx \mathbf{s}_{\boldsymbol{\theta}}(\mathbf{x})$.

To demonstrate the negative effect of this, we provide the result of a toy experiment (details in Appendix B.1) in Fig. 2 where we use sliced score matching to estimate scores of a mixture of Gaussians $p_{\text{data}} = \frac{1}{5}\mathcal{N}((-5, -5), I) + \frac{4}{5}\mathcal{N}((5, 5), I)$. As the figure demonstrates, score estimation is only reliable in the immediate vicinity of the modes of $p_{\text{data}}$, where the data density is high.

### 3.2.2 Slow mixing of Langevin dynamics

When two modes of the data distribution are separated by low density regions, Langevin dynamics will not be able to correctly recover the relative weights of these two modes in reasonable time, and therefore might not converge to the true distribution. Our analyses of this are largely inspired by [63], which analyzed the same phenomenon in the context of density estimation with score matching.

Consider a mixture distribution $p_{\text{data}}(\mathbf{x}) = \pi p_1(\mathbf{x}) + (1-\pi)p_2(\mathbf{x})$, where $p_1(\mathbf{x})$ and $p_2(\mathbf{x})$ are normalized distributions with disjoint supports, and $\pi \in (0, 1)$. In the support of $p_1(\mathbf{x})$, $\nabla_{\mathbf{x}}\log p_{\text{data}}(\mathbf{x}) = \nabla_{\mathbf{x}}(\log \pi + \log p_1(\mathbf{x})) = \nabla_{\mathbf{x}}\log p_1(\mathbf{x})$, and in the support of $p_2(\mathbf{x})$, $\nabla_{\mathbf{x}}\log p_{\text{data}}(\mathbf{x}) = \nabla_{\mathbf{x}}(\log(1-\pi) + \log p_2(\mathbf{x})) = \nabla_{\mathbf{x}}\log p_2(\mathbf{x})$. In either case, the score $\nabla_{\mathbf{x}}\log p_{\text{data}}(\mathbf{x})$ does not depend on $\pi$. Since Langevin dynamics use $\nabla_{\mathbf{x}}\log p_{\text{data}}(\mathbf{x})$ to sample from $p_{\text{data}}(\mathbf{x})$, the samples obtained will not depend on $\pi$. In practice, this analysis also holds when different modes have approximately disjoint supports—they may share the same support but be connected by regions of small data density. In this case, Langevin dynamics can produce correct samples in theory, but may require a very small step size and a very large number of steps to mix.

To verify this analysis, we test Langevin dynamics sampling for the same mixture of Gaussian used in Section 3.2.1 and provide the results in Fig. 3. We use the ground truth scores when sampling with Langevin dynamics. Comparing Fig. 3(b) with (a), it is obvious that the samples from Langevin dynamics have incorrect relative density between the two modes, as predicted by our analysis.

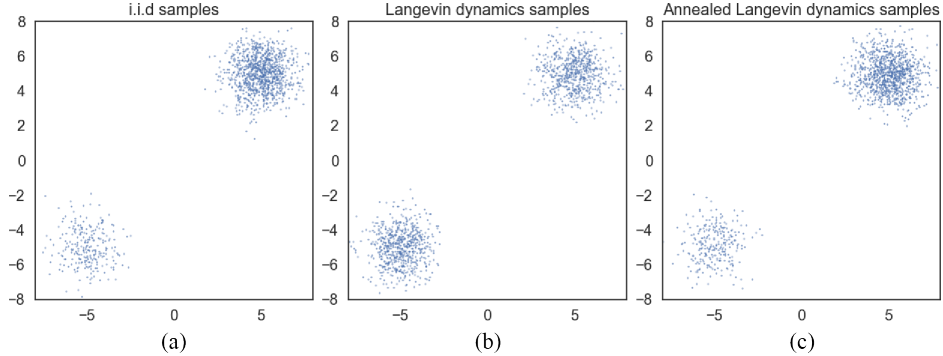

Figure 3: Samples from a mixture of Gaussian with different methods. (a) Exact sampling. (b) Sampling using Langevin dynamics with the exact scores. (c) Sampling using annealed Langevin dynamics with the exact scores. Clearly Langevin dynamics estimate the relative weights between the two modes incorrectly, while annealed Langevin dynamics recover the relative weights faithfully.

# 4 Noise Conditional Score Networks: learning and inference

We observe that perturbing data with random Gaussian noise makes the data distribution more amenable to score-based generative modeling. First, since the support of our Gaussian noise distribution is the whole space, the perturbed data will not be confined to a low dimensional manifold, which obviates difficulties from the manifold hypothesis and makes score estimation well-defined. Second, large Gaussian noise has the effect of filling low density regions in the original unperturbed data distribution; therefore score matching may get more training signal to improve score estimation. Furthermore, by using multiple noise levels we can obtain a sequence of noise-perturbed distributions that converge to the true data distribution. We can improve the mixing rate of Langevin dynamics on multimodal distributions by leveraging these intermediate distributions in the spirit of simulated annealing [30] and annealed importance sampling [37].

Built upon this intuition, we propose to improve score-based generative modeling by 1) *perturbing the data using various levels of noise*; and 2) *simultaneously estimating scores corresponding to all noise levels by training a single conditional score network*. After training, when using Langevin dynamics to generate samples, we initially use scores corresponding to large noise, and gradually anneal down the noise level. This helps smoothly transfer the benefits of large noise levels to low noise levels where the perturbed data are almost indistinguishable from the original ones. In what follows, we will elaborate more on the details of our method, including the architecture of our score networks, the training objective, and the annealing schedule for Langevin dynamics.

## 4.1 Noise Conditional Score Networks

Let $\{\sigma_i\}_{i=1}^L$ be a positive geometric sequence that satisfies $\frac{\sigma_1}{\sigma_2} = \cdots = \frac{\sigma_{L-1}}{\sigma_L} > 1$. Let $q_\sigma(\mathbf{x}) \triangleq \int p_{\text{data}}(\mathbf{t})\mathcal{N}(\mathbf{x} \mid \mathbf{t}, \sigma^2 I)\mathrm{d}\mathbf{t}$ denote the perturbed data distribution. We choose the noise levels $\{\sigma_i\}_{i=1}^L$ such that $\sigma_1$ is large enough to mitigate the difficulties discussed in Section 3, and $\sigma_L$ is small enough to minimize the effect on data. We aim to train a conditional score network to jointly estimate the scores of all perturbed data distributions, *i.e.*, $\forall \sigma \in \{\sigma_i\}_{i=1}^L : \mathbf{s}_\theta(\mathbf{x}, \sigma) \approx \nabla_{\mathbf{x}} \log q_\sigma(\mathbf{x})$. Note that $\mathbf{s}_\theta(\mathbf{x}, \sigma) \in \mathbb{R}^D$ when $\mathbf{x} \in \mathbb{R}^D$. We call $\mathbf{s}_\theta(\mathbf{x}, \sigma)$ a *Noise Conditional Score Network (NCSN)*.

Similar to likelihood-based generative models and GANs, the design of model architectures plays an important role in generating high quality samples. In this work, we mostly focus on architectures useful for image generation, and leave the architecture design for other domains as future work. Since the output of our noise conditional score network $\mathbf{s}_\theta(\mathbf{x}, \sigma)$ has the same shape as the input image $\mathbf{x}$, we draw inspiration from successful model architectures for dense prediction of images (*e.g.*, semantic segmentation). In the experiments, our model $\mathbf{s}_\theta(\mathbf{x}, \sigma)$ combines the architecture design of U-Net [46] with dilated/atrous convolution [64, 65, 8]—both of which have been proved very successful in semantic segmentation. In addition, we adopt instance normalization in our score network, inspired by its superior performance in some image generation tasks [57, 13, 23], and we

use a modified version of conditional instance normalization [13] to provide conditioning on $\sigma_i$. More details on our architecture can be found in Appendix A.

## 4.2 Learning NCSNs via score matching

Both sliced and denoising score matching can train NCSNs. We adopt denoising score matching as it is slightly faster and naturally fits the task of estimating scores of noise-perturbed data distributions. However, we emphasize that empirically sliced score matching can train NCSNs as well as denoising score matching. We choose the noise distribution to be $q_\sigma(\tilde{\mathbf{x}} \mid \mathbf{x}) = \mathcal{N}(\tilde{\mathbf{x}} \mid \mathbf{x}, \sigma^2 I)$; therefore $\nabla_{\tilde{\mathbf{x}}} \log q_\sigma(\tilde{\mathbf{x}} \mid \mathbf{x}) = -(\tilde{\mathbf{x}}-\mathbf{x})/\sigma^2$. For a given $\sigma$, the denoising score matching objective (Eq. (2)) is

$$\ell(\boldsymbol{\theta}; \sigma) \triangleq \frac{1}{2} \mathbb{E}_{p_{\text{data}}(\mathbf{x})} \mathbb{E}_{\tilde{\mathbf{x}} \sim \mathcal{N}(\mathbf{x}, \sigma^2 I)} \left[ \left\| \mathbf{s}_{\boldsymbol{\theta}}(\tilde{\mathbf{x}}, \sigma) + \frac{\tilde{\mathbf{x}} - \mathbf{x}}{\sigma^2} \right\|_2^2 \right]. \tag{5}$$

Then, we combine Eq. (5) for all $\sigma \in \{\sigma_i\}_{i=1}^L$ to get one unified objective

$$\mathcal{L}(\boldsymbol{\theta}; \{\sigma_i\}_{i=1}^L) \triangleq \frac{1}{L} \sum_{i=1}^L \lambda(\sigma_i) \ell(\boldsymbol{\theta}; \sigma_i), \tag{6}$$

where $\lambda(\sigma_i) > 0$ is a coefficient function depending on $\sigma_i$. Assuming $\mathbf{s}_{\boldsymbol{\theta}}(\mathbf{x}, \sigma)$ has enough capacity, $\mathbf{s}_{\boldsymbol{\theta}^*}(\mathbf{x}, \sigma)$ minimizes Eq. (6) if and only if $\mathbf{s}_{\boldsymbol{\theta}^*}(\mathbf{x}, \sigma_i) = \nabla_{\mathbf{x}} \log q_{\sigma_i}(\mathbf{x})$ a.s. for all $i \in \{1, 2, \cdots, L\}$, because Eq. (6) is a conical combination of $L$ denoising score matching objectives.

There can be many possible choices of $\lambda(\cdot)$. Ideally, we hope that the values of $\lambda(\sigma_i)\ell(\boldsymbol{\theta}; \sigma_i)$ for all $\{\sigma_i\}_{i=1}^L$ are roughly of the same order of magnitude. Empirically, we observe that when the score networks are trained to optimality, we approximately have $\|\mathbf{s}_{\boldsymbol{\theta}}(\mathbf{x}, \sigma)\|_2 \propto 1/\sigma$. This inspires us to choose $\lambda(\sigma) = \sigma^2$. Because under this choice, we have $\lambda(\sigma)\ell(\boldsymbol{\theta}; \sigma) = \sigma^2 \ell(\boldsymbol{\theta}; \sigma) = \frac{1}{2} \mathbb{E}[\|\sigma \mathbf{s}_{\boldsymbol{\theta}}(\tilde{\mathbf{x}}, \sigma) + \frac{\tilde{\mathbf{x}}-\mathbf{x}}{\sigma}\|_2^2]$. Since $\frac{\tilde{\mathbf{x}}-\mathbf{x}}{\sigma} \sim \mathcal{N}(0, I)$ and $\|\sigma \mathbf{s}_{\boldsymbol{\theta}}(\mathbf{x}, \sigma)\|_2 \propto 1$, we can easily conclude that the order of magnitude of $\lambda(\sigma)\ell(\boldsymbol{\theta}; \sigma)$ does not depend on $\sigma$.

We emphasize that our objective Eq. (6) requires no adversarial training, no surrogate losses, and no sampling from the score network during training (*e.g.*, unlike contrastive divergence). Also, it does not require $\mathbf{s}_{\boldsymbol{\theta}}(\mathbf{x}, \sigma)$ to have special architectures in order to be tractable. In addition, when $\lambda(\cdot)$ and $\{\sigma_i\}_{i=1}^L$ are fixed, it can be used to quantitatively compare different NCSNs.

## 4.3 NCSN inference via annealed Langevin dynamics

After the NCSN $\mathbf{s}_{\boldsymbol{\theta}}(\mathbf{x}, \sigma)$ is trained, we propose a sampling approach—annealed Langevin dynamics (Alg. 1)—to produced samples, inspired by simulated annealing [30] and annealed importance sampling [37]. As shown in Alg. 1, we start annealed Langevin dynamics by initializing the samples from some fixed prior distribution, *e.g.*, uniform noise. Then, we run Langevin dynamics to sample from $q_{\sigma_1}(\mathbf{x})$ with step size $\alpha_1$. Next, we run Langevin dynamics to sample from $q_{\sigma_2}(\mathbf{x})$, starting from the final samples of the previous simulation and using a reduced step size $\alpha_2$. We continue in this fashion, using the final samples of Langevin dynamics for $q_{\sigma_{i-1}}(\mathbf{x})$ as the initial samples of Langevin dynamic for

---

**Algorithm 1** Annealed Langevin dynamics.

**Require:** $\{\sigma_i\}_{i=1}^L, \epsilon, T$.
1: Initialize $\tilde{\mathbf{x}}_0$
2: **for** $i \leftarrow 1$ to $L$ **do**
3:      $\alpha_i \leftarrow \epsilon \cdot \sigma_i^2 / \sigma_L^2$     $\triangleright \alpha_i$ is the step size.
4:      **for** $t \leftarrow 1$ to $T$ **do**
5:          Draw $\mathbf{z}_t \sim \mathcal{N}(0, I)$
6:          $\tilde{\mathbf{x}}_t \leftarrow \tilde{\mathbf{x}}_{t-1} + \frac{\alpha_i}{2} \mathbf{s}_{\boldsymbol{\theta}}(\tilde{\mathbf{x}}_{t-1}, \sigma_i) + \sqrt{\alpha_i}\, \mathbf{z}_t$
7:      **end for**
8:      $\tilde{\mathbf{x}}_0 \leftarrow \tilde{\mathbf{x}}_T$
9: **end for**
    **return** $\tilde{\mathbf{x}}_T$

---

$q_{\sigma_i}(\mathbf{x})$, and tuning down the step size $\alpha_i$ gradually with $\alpha_i = \epsilon \cdot \sigma_i^2 / \sigma_L^2$. Finally, we run Langevin dynamics to sample from $q_{\sigma_L}(\mathbf{x})$, which is close to $p_{\text{data}}(\mathbf{x})$ when $\sigma_L \approx 0$.

Since the distributions $\{q_{\sigma_i}\}_{i=1}^L$ are all perturbed by Gaussian noise, their supports span the whole space and their scores are well-defined, avoiding difficulties from the manifold hypothesis. When $\sigma_1$ is sufficiently large, the low density regions of $q_{\sigma_1}(\mathbf{x})$ become small and the modes become less isolated. As discussed previously, this can make score estimation more accurate, and the mixing of Langevin dynamics faster. We can therefore assume that Langevin dynamics produce good samples for $q_{\sigma_1}(\mathbf{x})$. These samples are likely to come from high density regions of $q_{\sigma_1}(\mathbf{x})$, which means

| Model | Inception | FID |
|---|---|---|
| **CIFAR-10 Unconditional** | | |
| PixelCNN [59] | 4.60 | 65.93 |
| PixelIQN [42] | 5.29 | 49.46 |
| EBM [12] | 6.02 | 40.58 |
| WGAN-GP [18] | $7.86 \pm .07$ | 36.4 |
| MoLM [45] | $7.90 \pm .10$ | **18.9** |
| SNGAN [36] | $8.22 \pm .05$ | 21.7 |
| ProgressiveGAN [25] | $8.80 \pm .05$ | - |
| **NCSN (Ours)** | $\mathbf{8.87} \pm .12$ | 25.32 |
| **CIFAR-10 Conditional** | | |
| EBM [12] | 8.30 | 37.9 |
| SNGAN [36] | $8.60 \pm .08$ | 25.5 |
| BigGAN [6] | **9.22** | **14**.73 |

Table 1: Inception and FID scores for CIFAR-10

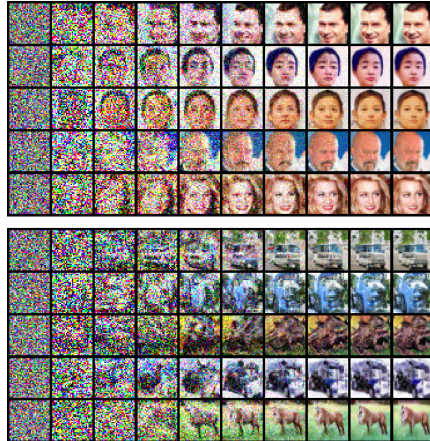

Figure 4: Intermediate samples of annealed Langevin dynamics.

they are also likely to reside in the high density regions of $q_{\sigma_2}(\mathbf{x})$, given that $q_{\sigma_1}(\mathbf{x})$ and $q_{\sigma_2}(\mathbf{x})$ only slightly differ from each other. As score estimation and Langevin dynamics perform better in high density regions, samples from $q_{\sigma_1}(\mathbf{x})$ will serve as good initial samples for Langevin dynamics of $q_{\sigma_2}(\mathbf{x})$. Similarly, $q_{\sigma_{i-1}}(\mathbf{x})$ provides good initial samples for $q_{\sigma_i}(\mathbf{x})$, and finally we obtain samples of good quality from $q_{\sigma_L}(\mathbf{x})$.

There could be many possible ways of tuning $\alpha_i$ according to $\sigma_i$ in Alg. 1. Our choice is $\alpha_i \propto \sigma_i^2$. The motivation is to fix the magnitude of the "signal-to-noise" ratio $\frac{\alpha_i \mathbf{s}_{\boldsymbol{\theta}}(\mathbf{x}, \sigma_i)}{2\sqrt{\alpha_i} \mathbf{z}}$ in Langevin dynamics. Note that $\mathbb{E}[\|\frac{\alpha_i \mathbf{s}_{\boldsymbol{\theta}}(\mathbf{x}, \sigma_i)}{2\sqrt{\alpha_i} \mathbf{z}}\|_2^2] \approx \mathbb{E}[\frac{\alpha_i \|\mathbf{s}_{\boldsymbol{\theta}}(\mathbf{x}, \sigma_i)\|_2^2}{4}] \propto \frac{1}{4}\mathbb{E}[\|\sigma_i \mathbf{s}_{\boldsymbol{\theta}}(\mathbf{x}, \sigma_i)\|_2^2]$. Recall that empirically we found $\|\mathbf{s}_{\boldsymbol{\theta}}(\mathbf{x}, \sigma)\|_2 \propto 1/\sigma$ when the score network is trained close to optimal, in which case $\mathbb{E}[\|\sigma_i \mathbf{s}_{\boldsymbol{\theta}}(\mathbf{x}; \sigma_i)\|_2^2] \propto 1$. Therefore $\|\frac{\alpha_i \mathbf{s}_{\boldsymbol{\theta}}(\mathbf{x}, \sigma_i)}{2\sqrt{\alpha_i} \mathbf{z}}\|_2 \propto \frac{1}{4}\mathbb{E}[\|\sigma_i \mathbf{s}_{\boldsymbol{\theta}}(\mathbf{x}, \sigma_i)\|_2^2] \propto \frac{1}{4}$ does not depend on $\sigma_i$.

To demonstrate the efficacy of our annealed Langevin dynamics, we provide a toy example where the goal is to sample from a mixture of Gaussian with two well-separated modes using only scores. We apply Alg. 1 to sample from the mixture of Gausssian used in Section 3.2. In the experiment, we choose $\{\sigma_i\}_{i=1}^L$ to be a geometric progression, with $L = 10$, $\sigma_1 = 10$ and $\sigma_{10} = 0.1$. The results are provided in Fig. 3. Comparing Fig. 3 (b) against (c), annealed Langevin dynamics correctly recover the relative weights between the two modes whereas standard Langevin dynamics fail.

## 5 Experiments

In this section, we demonstrate that our NCSNs are able to produce high quality image samples on several commonly used image datasets. In addition, we show that our models learn reasonable image representations by image inpainting experiments.

**Setup** We use MNIST, CelebA [34], and CIFAR-10 [31] datasets in our experiments. For CelebA, the images are first center-cropped to $140 \times 140$ and then resized to $32 \times 32$. All images are rescaled so that pixel values are in $[0, 1]$. We choose $L = 10$ different standard deviations such that $\{\sigma_i\}_{i=1}^L$ is a geometric sequence with $\sigma_1 = 1$ and $\sigma_{10} = 0.01$. Note that Gaussian noise of $\sigma = 0.01$ is almost indistinguishable to human eyes for image data. When using annealed Langevin dynamics for image generation, we choose $T = 100$ and $\epsilon = 2 \times 10^{-5}$, and use uniform noise as our initial samples. We found the results are robust w.r.t. the choice of $T$, and $\epsilon$ between $5 \times 10^{-6}$ and $5 \times 10^{-5}$ generally works fine. We provide additional details on model architecture and settings in Appendix A and B.

**Image generation** In Fig. 5, we show uncurated samples from annealed Langevin dynamics for MNIST, CelebA and CIFAR-10. As shown by the samples, our generated images have higher or comparable quality to those from modern likelihood-based models and GANs. To intuit the procedure of annealed Langevin dynamics, we provide intermediate samples in Fig. 4, where each row shows

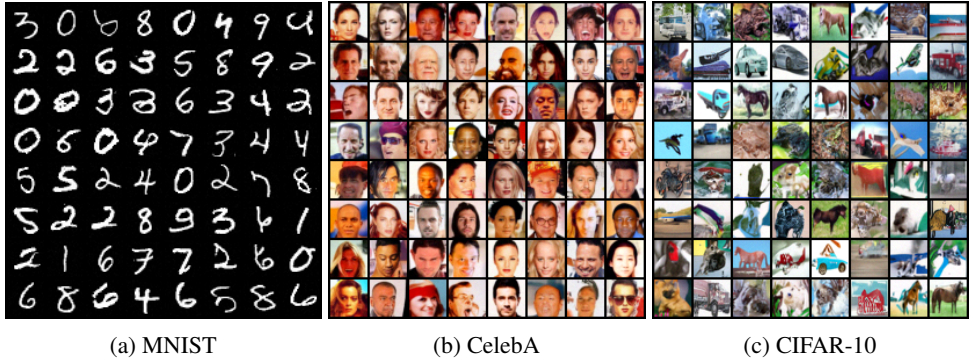

| (a) MNIST | (b) CelebA | (c) CIFAR-10 |

Figure 5: Uncurated samples on MNIST, CelebA, and CIFAR-10 datasets.

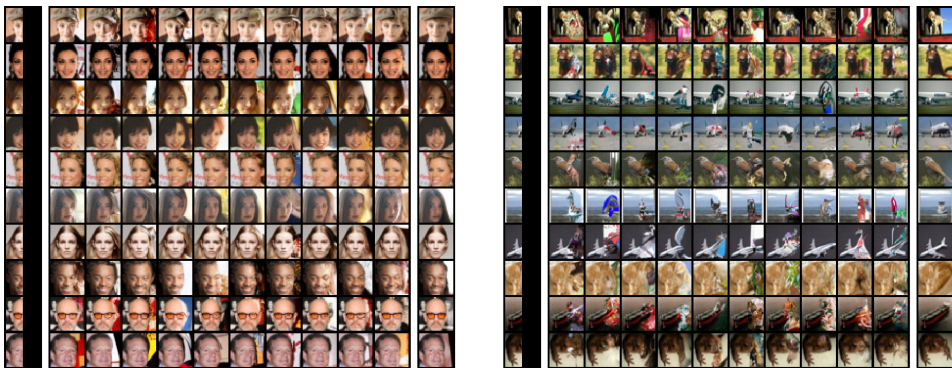

Figure 6: Image inpainting on CelebA (**left**) and CIFAR-10 (**right**). The leftmost column of each figure shows the occluded images, while the rightmost column shows the original images.

how samples evolve from pure random noise to high quality images. More samples from our approach can be found in Appendix C. We also show the nearest neighbors of generated images in the training dataset in Appendix C.2, in order to demonstrate that our model is not simply memorizing training images. To show it is important to learn a conditional score network jointly for many noise levels and use annealed Langevin dynamics, we compare against a baseline approach where we only consider one noise level $\{\sigma_1 = 0.01\}$ and use the vanilla Langevin dynamics sampling method. Although this small added noise helps circumvent the difficulty of the manifold hypothesis (as shown by Fig. 1, things will completely fail if no noise is added), it is not large enough to provide information on scores in regions of low data density. As a result, this baseline fails to generate reasonable images, as shown by samples in Appendix C.1.

For quantitative evaluation, we report inception [48] and FID [20] scores on CIFAR-10 in Tab. 1. As an *unconditional* model, we achieve the state-of-the-art inception score of 8.87, which is even better than most reported values for *class-conditional* generative models. Our FID score 25.32 on CIFAR-10 is also comparable to top existing models, such as SNGAN [36]. We omit scores on MNIST and CelebA as the scores on these two datasets are not widely reported, and different preprocessing (such as the center crop size of CelebA) can lead to numbers not directly comparable.

**Image inpainting**   In Fig. 6, we demonstrate that our score networks learn generalizable and semantically meaningful image representations that allow it to produce diverse image inpaintings. Note that some previous models such as PixelCNN can only impute images in the raster scan order. In contrast, our method can naturally handle images with occlusions of arbitrary shapes by a simple modification of the annealed Langevin dynamics procedure (details in Appendix B.3). We provide more image inpainting results in Appendix C.5.

# 6  Related work

Our approach has some similarities with methods that learn the transition operator of a Markov chain for sample generation [4, 51, 5, 16, 52]. For example, generative stochastic networks (GSN [4, 1]) use denoising autoencoders to train a Markov chain whose equilibrium distribution matches the data distribution. Similarly, our method trains the score function used in Langevin dynamics to sample from the data distribution. However, GSN often starts the chain very close to a training data point, and therefore requires the chain to transition quickly between different modes. In contrast, our annealed Langevin dynamics are initialized from unstructured noise. Nonequilibrium Thermodynamics (NET [51]) used a prescribed diffusion process to slowly transform data into random noise, and then learned to reverse this procedure by training an inverse diffusion. However, NET is not very scalable because it requires the diffusion process to have very small steps, and needs to simulate chains with thousands of steps at training time.

Previous approaches such as Infusion Training (IT [5]) and Variational Walkback (VW [16]) also employed different noise levels/temperatures for training transition operators of a Markov chain. Both IT and VW (as well as NET) train their models by maximizing the evidence lower bound of a suitable marginal likelihood. In practice, they tend to produce blurry image samples, similar to variational autoencoders. In contrast, our objective is based on score matching instead of likelihood, and we can produce images comparable to GANs.

There are several structural differences that further distinguish our approach from previous methods discussed above. First, *we do not need to sample from a Markov chain during training*. In contrast, the walkback procedure of GSNs needs multiple runs of the chain to generate "negative samples". Other methods including NET, IT, and VW also need to simulate a Markov chain for every input to compute the training loss. This difference makes our approach more efficient and scalable for training deep models. Secondly, *our training and sampling methods are decoupled from each other*. For score estimation, both sliced and denoising score matching can be used. For sampling, any method based on scores is applicable, including Langevin dynamics and (potentially) Hamiltonian Monte Carlo [38]. Our framework allows arbitrary combinations of score estimators and (gradient-based) sampling approaches, whereas most previous methods tie the model to a specific Markov chain. Finally, *our approach can be used to train energy-based models (EBM)* by using the gradient of an energy-based model as the score model. In contrast, it is unclear how previous methods that learn transition operators of Markov chains can be directly used for training EBMs.

Score matching was originally proposed for learning EBMs. However, many existing methods based on score matching are either not scalable [24] or fail to produce samples of comparable quality to VAEs or GANs [27, 49]. To obtain better performance on training deep energy-based models, some recent works have resorted to contrastive divergence [21], and propose to sample with Langevin dynamics for both training and testing [12, 39]. However, unlike our approach, contrastive divergence uses the computationally expensive procedure of Langevin dynamics as an inner loop during training. The idea of combining annealing with denoising score matching has also been investigated in previous work under different contexts. In [14, 7, 66], different annealing schedules on the noise for training denoising autoencoders are proposed. However, their work is on learning representations for improving the performance of classification, instead of generative modeling. The method of denoising score matching can also be derived from the perspective of Bayes least squares [43, 44], using techniques of Stein's Unbiased Risk Estimator [35, 56].

# 7  Conclusion

We propose the framework of score-based generative modeling where we first estimate gradients of data densities via score matching, and then generate samples via Langevin dynamics. We analyze several challenges faced by a naïve application of this approach, and propose to tackle them by training Noise Conditional Score Networks (NCSN) and sampling with annealed Langevin dynamics. Our approach requires no adversarial training, no MCMC sampling during training, and no special model architectures. Experimentally, we show that our approach can generate high quality images that were previously only produced by the best likelihood-based models and GANs. We achieve the new state-of-the-art inception score on CIFAR-10, and an FID score comparable to SNGANs.

## Acknowledgements

Toyota Research Institute ("TRI") provided funds to assist the authors with their research but this article solely reflects the opinions and conclusions of its authors and not TRI or any other Toyota entity. This research was also supported by NSF (#1651565, #1522054, #1733686), ONR (N00014-19-1-2145), AFOSR (FA9550-19-1-0024).

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
