[Supplementary Material · full.pdf]

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

# A  Architectures

The architecture of our NCSNs used in the experiments has three important components: instance normalization, dilated convolutions and U-Net-type architectures. Below we give more background on them and discuss how we modified them to suit our purpose. For more comprehensive details and a reference implementation, we recommend the readers to check our publicly available code base. Our score networks are implemented in `PyTorch`. Code and checkpoints are available at https://github.com/ermongroup/ncsn.

## A.1  Instance normalization

We use conditional instance normalization [13] so that $\mathbf{s}_{\boldsymbol{\theta}}(\mathbf{x}, \sigma)$ takes account of $\sigma$ when predicting the scores. In conditional instance normalization, a different set of scales and biases is used for different $\sigma \in \{\sigma_i\}_{i=1}^L$. More specifically, suppose $\mathbf{x}$ is an input with $C$ feature maps. Let $\mu_k$ and $s_k$ denote the mean and standard deviation of the $k$-th feature map of $\mathbf{x}$, taken along the spatial axes. Conditional instance normalization is achieved by

$$\mathbf{z}_k = \gamma[i, k]\frac{\mathbf{x}_k - \mu_k}{s_k} + \beta[i, k],$$

where $\gamma \in \mathbb{R}^{L \times C}$ and $\beta \in \mathbb{R}^{L \times C}$ are learnable parameters, $k$ denotes the index of feature maps, and $i$ denotes the index of $\sigma$ in $\{\sigma_i\}_{i=1}^L$.

However, one downside of instance normalization is that it completely removes the information of $\mu_k$ for different feature maps. This can lead to shifted colors in the generated images. To fix this issue, we propose a simple modification to conditional instance normalization. First, we compute the mean and standard deviation of $\mu_k$'s and denote them as $m$ and $v$ respectively. Then, we add another learnable parameter $\alpha \in \mathbb{R}^{L \times C}$. The modified conditional instance normalization is defined as

$$\mathbf{z}_k = \gamma[i, k]\frac{\mathbf{x}_k - \mu_k}{s_k} + \beta[i, k] + \alpha[i, k]\frac{\mu_k - m}{v}.$$

We abbreviate this modification of conditional instance normalization as CondInstanceNorm++. In our architecture, we add CondInstanceNorm++ before every convolutional layer and pooling layer.

## A.2  Dilated convolutions

Dilated convolutions can be used to increase the size of receptive field while maintaining the resolution of feature maps. It has been shown very effective in semantic segmentation because they preserve the location information better using feature maps of larger resolutions. In our architecture design of NCSNs, we use it to replace all the subsampling layers except the first one.

## A.3  U-Net architecture

U-Net is an architecture with special skip connections. These skip connections help transfer lower level information in shallow layers to deeper layers of the network. Since the shallower layers often contain low level information such as location and shape, these skip connections help improve the result of semantic segmentation. For building $\mathbf{s}_{\boldsymbol{\theta}}(\mathbf{x}, \sigma)$, we use the architecture of RefineNet [32], a modern variant of U-Net that also incorporates ResNet designs. We refer the readers to [32] for a detailed description of the RefineNet architecture.

In our experiments, we use a 4-cascaded RefineNet. We use pre-activation residual blocks. We remove all batch normalizations in the RefineNet architecture, and replace them with CondInstanceNorm++. We replace the max pooling layers in Refine Blocks with average pooling, as average pooling is reported to produce smoother images for image generation tasks such as style transfer. In addition, we also add CondInstanceNorm++ before each convolution and average pooling in the Refine Blocks, although no normalization is used in the original Refine Blocks. All activation functions are chosen to be ELU. As mentioned previously, we use dilated convolutions to replace the subsampling layers in residual blocks, except the first one. Following the common practice, we increase the dilation by a factor of 2 when proceeding to the next cascade. For CelebA and CIFAR-10 experiments, the number of filters for layers corresponding to the first cascade is 128, while the number of filters for other cascades are doubled. For MNIST experiments, the number of filters are halved.

# B    Additional experimental details

## B.1    Toy experiments

For the results in Fig. 1, we train a ResNet with sliced score matching on CIFAR-10. We use pre-activation residual blocks, and the ResNet is structured as an auto-encoder, where the encoder contains 5 residual blocks and the decoder mirrors the architecture of the encoder. The number of filters for each residual block of the encoder part is respectively 32, 64, 64, 128 and 128. The 2nd and 4th residual block of the encoder subsamples the feature maps by a factor of two. We use ELU activations throughout the network. We train the network with 50000 iterations using Adam optimizer and a batch size of 128 and learning rate of 0.001. The experiment was run on one Titan XP GPU.

For the results in Fig. 2, we choose $p_{\text{data}} = \frac{1}{5}\mathcal{N}((-5, -5), I) + \frac{4}{5}\mathcal{N}((5, 5), I)$. The score network is a 3-layer MLP with 128 hidden units and softplus activation functions. We train the score network with sliced score matching for 10000 iterations with Adam optimizer. The learning rate is 0.001, and the batch size is 128. The experiment was run on an Intel Core i7 GPU with 2.7GHz.

For the results in Fig. 3, we use the same toy distribution $p_{\text{data}} = \frac{1}{5}\mathcal{N}((-5, -5), I) + \frac{4}{5}\mathcal{N}((5, 5), I)$. We generate 1280 samples for each subfigure of Fig. 3. The initial samples are all uniformly chosen in the square $[-8, 8] \times [-8, 8]$. For Langevin dynamics, we use $T = 1000$ and $\epsilon = 0.1$. For annealed Langevin dynamics, we use $T = 100$, $L = 10$ and $\epsilon = 0.1$. We choose $\{\sigma_i\}_{i=1}^{L}$ to be a geometric progression, with $L = 10$, $\sigma_1 = 20$ and $\sigma_{10} = 1$. Both Langevin methods use the ground-truth data score for sampling. The experiment was run on an Intel Core i7 GPU with 2.7GHz.

## B.2    Image generation

During training, we randomly flip the images in CelebA and CIFAR-10. All models are optimized by Adam with learning rate 0.001 for a total of 200000 iterations. The batch size is fixed to 128. We save one checkpoint every 5000 iterations. For MNIST, we choose the last checkpoint at the 200000-th training iteration. For selecting our CIFAR-10 and CelebA models, we generate 1000 images for each checkpoint and choose the one with the smallest FID score computed on these 1000 images. Our image samples and results in Tab. 1 are from these checkpoints. Similar model selection procedures have been used in previous work, such as ProgressiveGAN [25].

The inception and FID scores are computed using the official code from OpenAI [1] [48] and TTUR [20] authors [2] respectively. The architectures are described in Appendix A. When reporting the numbers in Tab. 1, we compute inception and FID scores based on a total of 50000 samples.

The baseline model uses the same score network. The only difference is that the score network is only conditioned on one noise level $\{\sigma_1 = 0.01\}$. When sampling using Langevin dynamics, we use $\epsilon = 2 \times 10^{-5}$ and $T = 1000$.

The models on MNIST were run with one Titan XP GPU, while the models on CelebA and CIFAR-10 used two Titan XP GPUs.

## B.3    Image inpainting

We use the following Alg. 2 for image inpainting.

The hyperparameters are the same as those of the annealed Langevin dynamics used for image generation.

**Algorithm 2** Inpainting with annealed Langevin dynamics.

---

**Require:** $\{\sigma_i\}_{i=1}^L, \epsilon, T$    $\triangleright$ $\epsilon$ is smallest step size; $T$ is the number of iteration for each noise level.
**Require:** $\mathbf{m}, \mathbf{x}$          $\triangleright$ $\mathbf{m}$ is a mask to indicate regions not occluded; $\mathbf{x}$ is the given image.

1: Initialize $\tilde{\mathbf{x}}_0$
2: **for** $i \leftarrow 1$ to $L$ **do**
3:     $\alpha_i \leftarrow \epsilon \cdot \sigma_i^2 / \sigma_L^2$                                          $\triangleright$ $\alpha_i$ is the step size.
4:     Draw $\tilde{\mathbf{z}} \sim \mathcal{N}(0, \sigma_i^2)$
5:     $\mathbf{y} \leftarrow \mathbf{x} + \tilde{\mathbf{z}}$
6:     **for** $t \leftarrow 1$ to $T$ **do**
7:        Draw $\mathbf{z}_t \sim \mathcal{N}(0, I)$
8:        $\tilde{\mathbf{x}}_t \leftarrow \tilde{\mathbf{x}}_{t-1} + \dfrac{\alpha_i}{2} \mathbf{s}_{\boldsymbol{\theta}}(\tilde{\mathbf{x}}_{t-1}, \sigma_i) + \sqrt{\alpha_i}\, \mathbf{z}_t$
9:        $\tilde{\mathbf{x}}_t \leftarrow \tilde{\mathbf{x}}_t \odot (1 - \mathbf{m}) + \mathbf{y} \odot \mathbf{m}$
10:    **end for**
11:    $\tilde{\mathbf{x}}_0 \leftarrow \tilde{\mathbf{x}}_T$
12: **end for**
    **return** $\tilde{\mathbf{x}}_T$

---

## C  Samples

### C.1  Samples from the baseline models

(a) MNIST            (b) CelebA            (c) CIFAR-10

Figure 7: Uncurated samples on MNIST, CelebA, and CIFAR-10 datasets from the baseline model.

(a) MNIST            (b) CelebA            (c) CIFAR-10

Figure 8: Intermediate samples from Langevin dynamics for the baseline model.

## C.2 Nearest neighbors

Figure 9: Nearest neighbors measured by the $\ell_2$ distance between images. Images on the left of the red vertical line are samples from NCSN. Images on the right are nearest neighbors in the training dataset.

Figure 10: Nearest neighbors measured by the $\ell_2$ distance in the feature space of an Inception V3 network pretrained on ImageNet. Images on the left of the red vertical line are samples from NCSN. Images on the right are nearest neighbors in the training dataset.

## C.3 Extended samples

Figure 11: Extended MNIST samples

Figure 12: Extended CelebA samples

Figure 13: Extended CIFAR-10 samples

Figure 14: Extended intermediate samples from annealed Langevin dynamics for CelebA.

Figure 15: Extended intermediate samples from annealed Langevin dynamics for CelebA.

## C.5 Extended image inpainting results

Figure 16: Extended image inpainting results for CelebA. The leftmost column of each figure shows the occluded images, while the rightmost column shows the original images.

Figure 17: Extended image inpainting results for CIFAR-10. The leftmost column of each figure shows the occluded images, while the rightmost column shows the original images.