[Reviews · NeurIPS 2019]

Reviewer 1



============ Update: In response to the authors' response, I've raised my score to 8. ============ Significance: The paper achieves a generative model capable of generating samples comparable to GANs, without a difficult adversarial training process or massive amounts of compute and data. I think that the potential impact of such a thing is very large. Quality: The paper makes a careful analysis of problems with score matching for generative modeling. The claims are specific, and each claim is backed up with a carefully done toy experiment. The method is well-motivated and few choices seem ad-hoc. The image generation experiments aren't fantastic, but they're more than sufficient to support the claim that the model performs comparably to GANs and better than any other non-GAN-related method I've seen (I'm counting MoLM and https://arxiv.org/abs/1903.08689 both as GAN-related methods). Originality: I don't have a deep background on score matching or energy models, but to my knowledge the ideas presented are novel. Even if they weren't, I would argue that putting them together into an algorithm which achieves competitive performance is a novel contribution in its own right. Clarity: The paper is well-written, with enough background and detail to be easily understood. I do have substantial criticisms which I think should be addressed before this paper is published (see the 'improvements' section of the review), but if these are addressed I recommend acceptance.

Reviewer 2



UPDATE: the authors have taken care to address all of my concerns in their rebuttal. Please ensure that they are also incorporated in the camera-ready version of the paper. The experiments on CelebA 64x64 are a welcome addition, although I would argue that this dataset isn't necessarily more challenging: the higher resolution is compensated for by simpler image content (carefully aligned faces). The fact that the cost of sampling does not increase is very promising though. I look forward to seeing this method being scaled up further to the level of adversarial and likelihood-based models in the future. I would also encourage the authors to try out varying the noise level continuously. The model could be conditioned on the logarithm of the variance of the noise, for example. In light of the quality of the rebuttal I have raised my score further. --------- The paper describes an alternative paradigm for generative modelling using score function estimation. Although several previous works have used estimators based on score functions to train generative models, these models usually try to capture the (unnormalised) density. Here, the score function is modelled directly instead. Sampling from such a model can be done using Langevin dynamics [47]. As far as I know, previous attempts at modelling score functions directly have not been successful. It is worth noting that Saremi et al. attempted something similar in "Deep Energy Estimator Networks" (also co-authored by Aapo Hyvarinen), but report that their experiments failed (top of page 7 in the arxiv version of their paper). I think it constitutes relevant related work and I would appreciate a comment from the authors regarding Saremi et al.'s discussion about modelling the score function vs. the energy function, as they seem to imply that modelling the score function directly is not feasible, whereas this work clearly demonstrates that it is. The authors observe that score estimation is difficult in low-density regions, which poses a problem for sampling through Langevin dynamics, which is likely to have low-density starting point. They propose adding random noise to the data and training a single noise-level-conditional model that can capture the score at different levels of noise. In some sense, this parameter sharing across noise levels allows the score estimates at different noise levels to regularise each other. I think this is a clever strategy to tackle this issue: although it means that we now need to anneal the noise level during sampling, the Langevin dynamics sampling procedure was already iterative anyway, so this doesn't actually complicate matters in terms of computational expense. One thing was not entirely clear to me though: in practice, a number of discrete noise levels are chosen, and the model is only trained for those noise levels (and not the levels in between). Since this is a continuous parameter, why not vary it continuously, at least during training? It would be helpful to clarify the motivation for the discrete strategy in the paper as well. Apart from this one point, the paper is very clearly written and easy to follow. Other comments: - line 33: the statement "we explore a new principle for generative modelling" confused me a bit as I was thinking of score matching, which arguably isn't new. More clearly distinguishing the proposed method from previous uses of score matching for generative modelling in literature could clarify things at this point. - line 115: this mention of "covariant derivatives" was also confusing, and I had to go look up what they actually are. It could be helpful to add a very brief explanation in a footnote or in parentheses, or just remove the mention altogether as the concept is not used in the rest of the paper. - line 237: I don't think "higher fidelity" is appropriate or justifiable here, and "comparable fidelity" suffices. The samples definitely don't look noticeably better than some that I've seen before from adversarial / likelihood-based models. If the authors want to make this point, it would be better to compare with samples from state-of-the-art models side by side in the figures. - line 243: this is a really nice result, as it demonstrates the necessity and effectiveness of the proposed noise-conditional modelling approach. It is unfortunate that it has been relegated to the appendix, but I understand that space constraints probably led to this decision. If there is room in the camera-ready version, I think it would be great to have this result in the main paper. - line 265: This statement could be a bit misleading as both likelihood-based and adversarial models have been scaled up far beyond 32x32 images, and "high-fidelity" tends to imply higher resolutions than this.

Reviewer 3



the paper discusses a new learning principle of score-matching in the context of generative models. while score-matching is a pretty classical idea, the paper nicely demonstrates its power on large scale generative models and addresses the unique challenges posed up modern images. I have only one major comment. it is nice to see how score matching is able to generate non-blurry image examples. however, it came as unintuitive specifically because the learning algorithm adds gaussian noise --- gaussian noise in VAE is known to be the culprit of image blurriness. on the other hand, https://arxiv.org/abs/1903.05789 has shown that if we get the dimensionality of the manifold correct, we are also able to get non-blurry images. so I wonder how to reconcile the intuition of non-blurry generated images in the presence of gaussian noise? or is the generated images non-blurry because it gets the manifold dimension correct? if so, is the gain of the proposed algorithm coming from the new learning principle or getting the manifold dimension correct?

[Author Response · NeurIPS 2019]

We thank all the reviewers for providing valuable feedback. In what follows, we address specific questions.

**Q1 (R2):** *Motivation for parameterizing the score function explicitly, rather than as the gradient of an energy model.*

The main motivation is computational. When using the gradient of an energy model as the score network, we first need one backpropagation to compute the score, and then another backpropagation to optimize the parameters. This requires higher-order gradients, which has no or limited support in many deep learning frameworks (*e.g.*, mxnet). Also, this makes computation 4 to 9 times slower due to double backpropagation (*cf.*, this Github issue), compared to directly parameterizing the score with a similar architecture. We will discuss this motivation in Section 2.1.

**Q2 (R2):** *Metrics or experiments to assess whether the model is overfitting or memorizing the dataset.*

Our experiment on image inpainting (Figure 5) already shows that the model is not memorizing, since we are able to generate diverse reconstructions that are different from the original unoccluded image from the datatset. As suggested by R2, we will include training/test learning curves and nearest neighbors in the appendix. For example, we provide a subset of images in the following, where the left shows the curves on CelebA, and the right shows the top 10 nearest neighbors of two samples (to the left of the red line) in pixel space (top two rows) and feature space of an Inception V3 network (bottom two rows). As expected, our model is not overfitting or memorizing the dataset.

**Q3 (R2):** *Issues on CIFAR-10 inception scores.*

When computing inception scores, we did not flip images. Flipping was only done in training, which is a common data augmentation technique used in training other generative models as well (*e.g.*, i-ResNet). We will clarify this more in Appendix C.2. The numbers in Table 1 are mainly from Figure 5 of the OpenAI EBM paper (arxiv: 1903.08689v1). We agree with R2 that the inception score of WGAN-GP should be 7.86, and will correct this in the paper. We double checked other numbers in the table, and they matched numbers reported in previous work. Our inception score was computed using the original code from OpenAI, and FID score was computed with the original code of TTUR authors.

**Q4 (R2, R3):** *More discussion on related work.*

Since NeurIPS allows an additional page for camera ready, we will include a more comprehensive related work section if our paper is accepted. We will discuss Deep Energy Estimator Networks (suggsted by R3), and move the related work part in the conclusion section to this section (suggested by R2).

**Q5 (R3):** *Why not use continuously varying noise levels? Writing suggestions on line 33, 115, 237, 243, and 265.*

The discrete strategy is a design choice to simplify the implementation. This makes it easier for us to borrow architectural designs from existing models (we are aware of more models conditioned on discrete labels than continuous ones). We appreciate all the writing suggestions. They are very helpful and we will incorporate them in the paper.

**Q6 (R3):** *Scalability to higher resolution images.*

We tried modeling $64 \times 64$ CelebA images. We empirically found that our models can be scaled to higher resolution images, and the cost of sampling is the same to $32 \times 32$ images, in terms of using the same number of iterations. We will incorporate these results into the paper. Some uncurated samples are provided below.

**Q7 (R3):** *Extension of score-based generative modeling to discrete data.*

There are many extensions of score matching to discrete data, *e.g.*, ratio matching and minimum probability flows. We could possibly couple them with appropriate MCMC sampling methods and annealing strategies.

**Q8 (R4):** *Why learning with score-matching can avoid the blurriness of samples in the presence of Gaussian noise.*

During sampling, we decrease the variance of the Gaussian noise. The final variance is 0.0001, and a Gaussian perturbation with this variance is almost indistinguishable (to human eyes) when the pixel values are within $[0, 1]$. Therefore, using Gaussian noise in our model does not necessarily mean generated images should be blurry.

[Meta-Review · NeurIPS 2019]

The paper proposes to perform Langevin dynamics in data space (as opposed to the latent space) of a deep generative model as a means to explore the data distribution. This reduces the difficult problem of estimating the data distribution to the slightly less difficult problem of estimating its gradients. The latter ones are estimated by different versions of score matching. This paper mainly builds on recent work on score matching by random projections. As a result, a new generative model is achieved whose sample quality is similar to GANs, while avoiding an adversarial training paradigm. This is a strong contribution. As a minor point of criticism, the reviewers wished a more thorough analysis of the effect of extra noise added on the data, which could be provided for the camera-ready version.